

# Childhood parasitic infections and gastrointestinal illness in indigenous communities at Lake Atitlán, Guatemala

Amber Roegner[1,2], Mónica N. Orozco[3], Claudia Jarquin[4], William Boegel[5], Clara Secaira[6], Marlin E. Caballeros[7], Lujain Al-Saleh[8] and Eliška Rejmánková[2]

[1] Center For Global Health, University of Oregon, Eugene, OR, United States of America
[2] Department of Environmental Science and Policy, University of California, Davis, Davis, CA, United States of America
[3] Center for Atitlán Studies, Universidad del Valle de Guatemala, Sololá, Sololá, Guatemala
[4] Center for Health Studies, Universidad del Valle de Guatemala, Guatemala City, Guatemala
[5] Opal House, Agua Escondida, Sololá, Guatemala
[6] Sololá, Sololá, Guatemala
[7] Laboratorio La Asunción, Sololá, Sololá, Guatemala
[8] School of Public Health, University of California, Berkeley, Berkeley, CA, United States of America

Corresponding author
Amber Roegner,
afroegner@gmail.com

## ABSTRACT

Lake Atitlán has experienced a decline in water quality resulting from cultural eutrophication. Indigenous Mayans who already face disproportionate health challenges rely directly on the lake water. Our objectives were to: (1) estimate prevalence of shedding of water-borne fecal parasites among children 5 years of age and younger, (2) assess household-reported incidence of gastrointestinal illness in children, and (3) characterize water sources, treatment, and sanitation conditions in households. We hypothesized that household use of untreated lake water results in increased risk of shedding of parasites and gastrointestinal symptoms. A cross-sectional fecal sampling and physical exam of 401 children were conducted along with WASH surveys in partnership with healthcare providers in seven communities. Fecal samples were screened for *Giardia lamblia* and *Cryptosporidium parvum,* using a rapid ELISA, with a portion examined by microscope. The prevalence of parasite shedding was 12.2% (9.7% for *Giardia*; 2.5% for *Cryptosporidium*). Risk factors for *Giardia* shedding included age 3 years or older (3.4 odds ratio, z-stat = 2.781 $p = 0.0054$), low height-for-age z-score (2.3 odds ratio, z-stat = 2.225, $p = 0.0216$), lack of any household water treatment (2.5 odds ratio, z-stat = 2.492, $p < 0.0012$), and open access to household latrine (2.04 odds ratio, z-stat = 1.992, $p = 0.0464$). The majority (77.3%) of households reported water treatment, boiling and gravity fed filters as the most widespread practices. The vast majority of households (92%) reported usage of a latrine, while 40% reported open and shared access beyond their household. An overwhelming majority of households reported diarrhea and fever several times per year or greater, with approximately half reporting vomiting at that frequency. Lake water use was identified as a risk factor for households reporting frequent gastrointestinal symptoms (odds ratio of 2.5, 4.4, and 2.6; z-stat of 3.10, 3.65, and 3.0; *p*-values of 0.0021, 0.0003, and 0.0028, for diarrhea, vomiting, and fever, respectively) in children 5 years of age and younger. The frequency of gastrointestinal illness with a strong link to lake drinking water cannot be explained by the prevalence of protozoa, and risk from other enteropathogens must be explored.

> Improving access to water treatment and sanitation practices could substantially reduce the parasite burden faced by developing children in the region.

# INTRODUCTION

The physical, political and sociocultural geography of our study area is complex and integral to understanding the access and barriers to clean drinking water. A crater lake situated in the Western Highlands of the Sierra Madre range in Guatemala, Atitlán spans approximately 125 km$^2$ between three volcanoes at an elevation of 1,562 m, and is greater than 300 m deep (*Brezonik & Fox, 1974*; *Newhall et al., 1987*). Described as "the most beautiful lake in the world," Atitlán continues to attract international tourists, expatriates, and internal migration. This tourism and human movement, compounded by a lack of urban and territorial planning, has resulted in disorderly population growth, expansion of agricultural monoculture, deforestation through slash and burn activities, and this recent change has occurred with little to no wastewater management (*Castellanos et al., 2002*; *Ferráns et al., 2018*; *Roegner et al., 2016*). Streams and runoff from an expansive watershed (540 km$^2$) drain high nutrient and contaminant loads into the lake, with direct sewage inflow (*Castellanos et al., 2002*; *Corman et al., 2015*; *Ferráns et al., 2018*; *Rejmánková et al., 2011*).

This deep, once oligotrophic lake, serves as the primary drinking water source for more than 70,000 inhabitants around the lake (*Orozco, 2017*; *Roegner et al., 2016*). It provides ecosystem services for a basin of close to a $\frac{1}{2}$ million people. The twelve municipalities on the lake shores face dilapidated and undersized, or completely lack, wastewater treatment facilities (*Chandra et al., 2013*). The persistent entry of raw sewage into the lake, estimated at 45,000 m$^3$ per day, has resulted in direct fecal contamination (*Ferráns et al., 2018*; *Orozco, 2017*; *Roegner et al., 2016*). At least five lakeside towns draw a portion of municipal water directly from the lake. Chlorination is mandated by government code, although enforcement is absent, and previous anthropological work has suggested variable distrust of chlorination among the Mayan populations (*Nagata et al., 2011a*; *Nagata et al., 2011b*; *Orozco, 2017*). In lake waters, there is repeated evidence of degraded water quality with elevated coliforms and *E. coli* at intake, point sources, and nonpoint sources (*Castellanos et al., 2002*), and 24% of monitored drinking water facilities, although many go unmonitored, deliver unacceptable levels of bacterial contamination (*Ferráns et al., 2018*; *INE, 2018*; *Orozco, 2017*).

The indigenous population in Guatemala is heterogenous with three dominant ethnicities in the Sololá Department, K'iche', Kaqchikel, and Tz'utujil, the latter two predominant in the basin. They face disproportionate health risks compared to the non-indigenous (*Chen et al., 2017*; *Jensen et al., 2009*; *Nagata et al., 2011a*; *Nagata et al., 2011b*; *Nagata et al., 2009*). Worldwide, indigenous communities face increased health risks

alongside lack of access to health care and poverty (*MSPAS, 2019*; *Omarova et al., 2018*; *Poder & He, 2015*). Over 70% of the indigenous population in the Sololá Department live in extreme poverty (*Omarova et al., 2018*; *Poder & He, 2015*). They constitute the majority of individuals directly utilizing lake water in the region. Within the department of Sololá, there were 28,041 and 32,724 severe diarrheal cases in children <5 years of age (rate of 5251.47 and 5967.79 per 100,000 inhabitants) during 2018 and 2019, respectively (*MSPAS, 2019*). There have been few studies looking at prevalence of waterborne illness in the region, although 11.2 deaths per 100,000 inhabitants in Sololá department and nationwide in Guatemala were related to gastroenteritis and other infectious colitis in 2017 (*INE, 2020*; *INE, 2020*). One study in 2004 found that 32% of residents from two villages at Atitlán had *Cryptosporidium parvum* infections, with the highest prevalence among female children and increased risk among those with a contaminated water supply from the lake (*Laubach et al., 2014*). Abdominal pain and diarrhea in school age children between ages 2 to 13 has been strongly correlated with shedding of *Cryptosporidium* oocysts in the region (*Bentley et al., 2004*).

Intestinal protozoa, such as *Cryptosporidium* and *Giardia*, cause symptomatic diarrhea and represent a frequently neglected public health risk with both among leading causes of moderate to severe diarrhea in children (*Bentley et al., 2004*).They also threaten development through prolonged inflammation and oxidative stress in the gut lining, particularly for infants and developing children, even in asymptomatic individuals (*Certad et al., 2017*; *Mmbaga & Houpt, 2017*). Prolonged disruption of the gut-lining results in inflammatory bowel disease and malabsorption. By substantially altering gut health, dietary intake and nutrients available to pregnant or nursing mother and child, it also directly impacts growth and cognitive development (*Certad et al., 2017*; *Mmbaga & Houpt, 2017*; *Mohammadkhah et al., 2018*). Guatemala has one of the highest levels of stunting among children under 5 years of age in the world, close to 50%. Previous work in the region has linked malabsorption and evidence of systemic oxidative stress (*Solomons et al., 2015*; *Soto-Méndez et al., 2017*; *Soto-Méndez et al., 2016*), as well as parasitic intestinal infections, to retarded growth curves in infants and preschool aged children (*Farthing et al., 1986*). Indeed, a retrospective study of 10,586 children of ages 5–15 in Guatemala over a four-year period (2004–2007) found increased risk of severe malnutrition with infection with *Giardia lamblia* and *Entamoeba histolytica* (*Jensen et al., 2009*). Systemic and urinary biomarkers of oxidative stress in preschoolers in the western highlands have been strongly correlated with *Giardia* infections (*Soto-Méndez et al., 2016*). Thus, not only are intestinal protozoa of immediate concern for symptomatic diarrhea and public health, but also for the long-term developmental consequences for children in the region.

Given chronic health implications of protozoal infection in young children, the raw sewage going directly in the lake, and multifactorial health risks and disproportionate poverty in indigenous populations, we aimed to examine linkages between indigenous household water sources and water, sanitation, hygiene (WASH) practices, gastrointestinal illness, and fecal protozoal parasites. Our objectives were to: (1) estimate prevalence of shedding of most common fecal parasites of human origin (*Giardia lamblia*, *Cryptosporidium parvum* and *Entamoeba histolytica*) among children 5 years of age and

younger in communities around Lake Atitlán, (2) assess household-reported incidence of gastrointestinal illness in those children, and (3) characterize water sources, any treatment of water, and sanitation conditions for households with children 5 years old and younger. We hypothesized that household use of untreated lake water would result in increased risk of shedding of parasites and household-reported disease.

## MATERIALS & METHODS

### Overall study design

To determine the prevalence of fecal parasite shedding in children under 5 years of age and obtain information on water sources, household treatment, and sanitation conditions in the home, a cross-sectional fecal sampling and WASH survey of 401 children was conducted in partnership with local NGOs and participating local physicians and health care providers (*see Acknowledgements*). With a goal of 400 subjects distributed equally between spring and lake water sources, subjects were recruited from seven communities around the lake, four of which were known to draw the majority or a portion of municipal water from the lake directly (San Lucas Tolimán, Santiago Atitlán, San Pedro La Laguna, and Panajachel), and three with mountain-derived spring water sources San Jorge de La Laguna, Tierra Linda, and Agua Escondida). Fecal samples were kept refrigerated, collected within 24 h of study participation, and analyzed within 24 h.

### Subject recruitment & consent

In September through November 2017, families with children 5 years old or younger were recruited to participate. Recruitment occurred through clinics, nurseries, and primary health providers in the region, as a part of routine check-ups, drop-in services, vaccination clinics, or group health and educational activities. Human subject approval and work with sensitive subjects (children) was approved through the Institutional Review Board at University of California, Davis (IRB # 1070493-3) and through the Comité de Ética en Investigación of the Centro de Estudios en Salud (CEI-CES) at Universidad del Valle de Guatemala (No. Protocol 167-07-2017).

Parents or guardians were asked to provide written informed consent, translated to Spanish or local language, before participating in any study activities. Prior to the day of participation, parents or guardians were given a sterile container for collection of feces from children, which they were instructed to collect and return within 24 h to the clinic to initiate participation in the study. To compensate study participation, parents were provided with a free bag of the nutritional, plant-based protein-rich beverage for infants and young children (*Incaparina*®). They also received educational training in how to reduce risks for their children and others in their family from water borne diseases, including information on water treatment options (Appendix A). In addition, where applicable, positive fecal test results were relayed to health care providers within one week and delivered to study participants, along with the appropriate medication. Appendix B details the medication treatment regimen included in the study.

## Questionnaire, physical exam, and sample collection

Single visit study participation included a WASH questionnaire (Appendix C) administered to parent or guardian by study personnel, a physical exam performed by the facility physician with data noted by multilingual assistant (Appendix D), and submission of collected stool samples for laboratory analysis. Prior to initiation of visits, all participating physicians and study personnel underwent training for human subjects and confidentiality, as well as a thorough overview of protocol, study objectives, and design. Inclusion criteria included any children ages 5 years of age or younger living in lakeside communities within the department of Sololá, who sought care at participating clinics and whose parents or guardians provided informed written consent. Exclusion criteria included age older than 5 years, acute sickness beyond possible physical examination as determined by attending physician, absence of consent from parent/guardian or child, unwillingness to provide a stool sample, or children suffering from confounding allergic diseases, such as irritable bowel disease, Crohn's disease, food allergies, and Celiac disease, per previous clinical record or diagnosis. Children were not excluded nor actively included based on presence or absence of episodic cases of gastrointestinal symptoms if no specific diagnoses had been made by their healthcare provider.

## Collection of stool sample and transport

During recruitment, the participant's parent or guardian was provided with sterile gloves and a specimen collection container, and instructions on how to collect a stool sample. Submitted fecal specimens were placed inside a plastic bag and samples in coolers were transported within hours to Laboratorio La Asunción, Sololá. Fecal samples were only analyzed if they were fresh samples or collected within the last 24 h with community healthcare liaison and health care providers verifying timeline. If fecal samples were not collected within 24 h, the patient data was excluded from the study.

## Fecal laboratory analysis

The samples were screened for *Giardia and Cryptosporidium* using the GIARDIA/CRYP-TOSPORIDIUM CHEK® (heretofore, 'Chek') (sensitivity 98.4%, specificity 100%, by independent study) (*Youn et al., 2009*) according to manufacturer instructions (TECHLAB 2016, ALERE, Blacksburg, VA). The ELISA method relies on monoclonal and polyclonal antibodies to detect cell–surface antigens of the *Giardia* cyst or *Cryptosporidium* oocyst in human fecal samples. Samples were brought to room temperature and diluted with diluent at a 4:1 ratio (400 µl of diluent and 100 µl or 0.1 gram of feces), in a microcentrifuge tube, then mixed thoroughly with a vortex. After the addition of 100 µl in each well of the microtiter plate, 50 µl of control or sample was added to each well; following a series of incubation and washing step, the samples were read in an ELISA plate reader at 450 nm or 450/620 nm.

For a subset of samples (140/401), the TRIAGE MICROPARASITE PANEL (heretofore, 'Triage') (ALERE, Blacksburg, VA) was used for simultaneous detection of *Giardia lamblia, Cryptosporidium parvum,* and *Entamoeba histolytica/dispar,* following manufacturer's instructions. Briefly, the method consists of a color change reaction that involves the

application of 500 µl of microcentrifuged filtered sample, from an initial 500 µl fecal samples dissolved in 4.5 ml diluent, to a panel, followed by addition of a conjugate, incubation step, a washing step, and addition of a substrate, read after five minutes. Provided all three control lines appeared within the test zone, a color bar of any intensity appearing with the sample test zones, indicates presence of those organisms. Additionally, a randomly selected subset (268/401) were screened by laboratory personnel through the microscope, after performing a fecal float and smear (*Broussard, 2003*).

## Data analysis

Raw data and specimens were archived only for the duration of the study. Analysis of anonymized and coded survey and physical examination data was undertaken with the assistance of RStudio® opensource software (*RStudio Team, 2017*). Z-scores and age-weighted BMIs were calculated according to World Health Organization guidelines. Descriptive statistics were used initially to characterize the physical exam data and WASH questionnaire, as well as to describe the prevalence of shedding of fecal parasites overall, and later relative to gender, community, water sources, and $z$-scores and WHO-adjusted BMI. Student $t$-tests and an ANOVA were used to look for potential differences between genders and communities, respectively.

In addition, logistic regression and chi square analysis of various categorical variables, utilizing a 95% confidence interval and with a $p$-value significance of less than or equal to 0.05, with respect to adverse health outcome (positive fecal sample or frequently reported symptoms) were systematically carried out to test various hypotheses relative to household water source, treatment, hygiene, and sanitation practices. Subsequently, for identified risk factors, odds ratios were determined for each participant characteristics and household practices or outcomes of protozoa shedding or gastrointestinal symptoms.

# RESULTS

## Descriptive characteristics of study participants

A total of 401 participants were enrolled in the study, with 215 males and 186 females after exclusion and inclusion criteria were applied. Figure 1 illustrates distribution of male and female participants at each site around the lake. San Lucas Tolimán, Santiago Atitlán, and San Pedro La Laguna were selected as sites with potential for lake sourced water within households, while San Jorge de La Laguna, Tierra Linda, and Agua Escondida were selected as sites with exclusively mountain spring water sources, although one household did report lake water usage at Tierra Linda (3.6%). Panajachel was thought to be predominantly spring water sourced, as well, however some participants did report lake water use ($n = 4$, 13.3%), possibly because of proximity to the lake shores compared with the other spring sources. In contrast, far fewer households in San Lucas Tolimán ($n = 44$, 35.8%), Santiago Atitlán ($n = 9$, 26.5%), and San Pedro La Laguna ($n = 1$, 6.7%) reported lake water use. To account for this discrepancy, we relied on household reporting of water source instead of physical location for all households. Thus, the final number of households reporting lake water use was 62, while 339 households reported an alternate source.
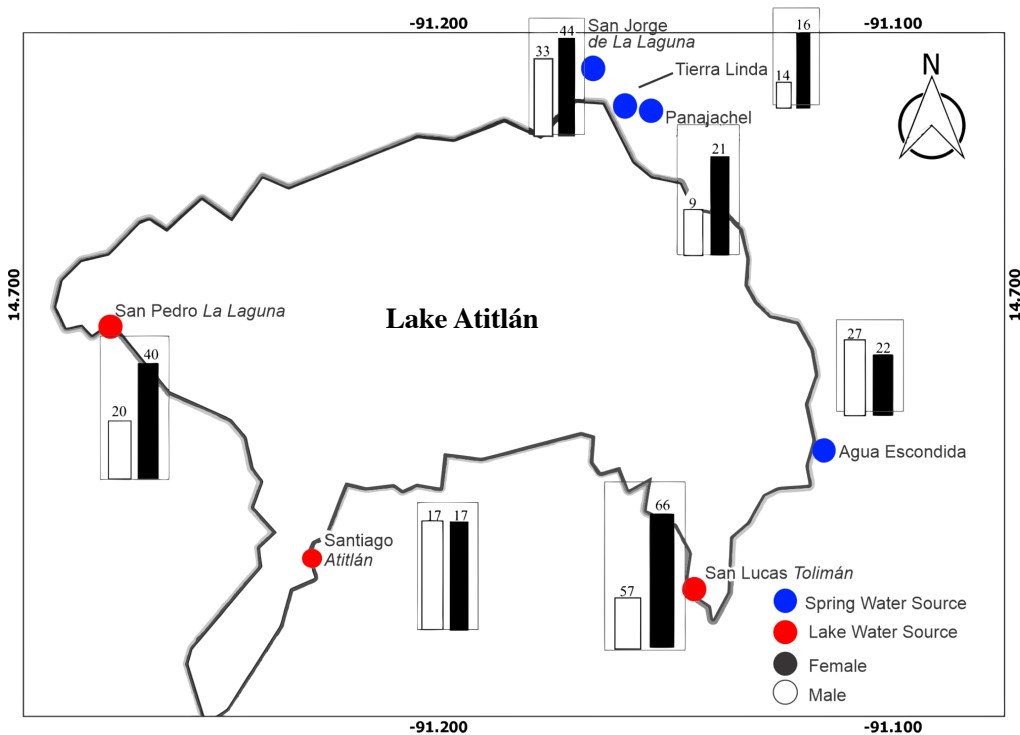

**Figure 1** **Map of study area with distribution of male and female participants at each community location is depicted.** There were a total of 215 and 186 male and female participants, respectively. The bar graphs depict number of male and female participants in black and white, respectively. Blue dots represent towns thought to predominantly derive water from spring sources, while red dots represent towns with a portion of water drawn from the lake prior to dispersal to households.

Overall, a greater number of participants were recruited from lake sourced towns (53.4%) as compared with spring sourced towns (46.6%), in anticipation of a proportion of the lake sourced town households actually having spring sources. The mean age of participants was $34.2 \pm 16.9$ months with a median of 36.0 months- girls were slightly older at $35 \pm 16.5$ months and boys at $33.5 \pm 17.4$.

There was a significant difference of BMI between male and female children (t-statistic $= -2.4413$, $p = 0.015$) with males having an increased BMI ($17.0 \pm 3.2$ kg/m$^2$) relative to females ($16.2 \pm 3.3$ kg/m$^2$) (Table S1). Overall the BMI is above the WHO mean for both genders for the mean age of the study ($35 \pm 16.4$ months and $33.5 \pm 17.5$ months, for males and females, respectively). There was significant variation across communities (Table S2) in height, weight, and BMI, mirroring the significant variation in age and distribution of gender across communities. As we relied on healthcare professionals serving different target populations at their point of service, recruitment by age and sex was highly variable (Fig. S1). However, Figs. 2A–2D illustrates the WHO (age-weighted) $z$-score for BMI (A, B) and height-for-age (C, D) divided according to gender, with a lower than average profile for height-for-age and above average distribution for age weighted BMI.
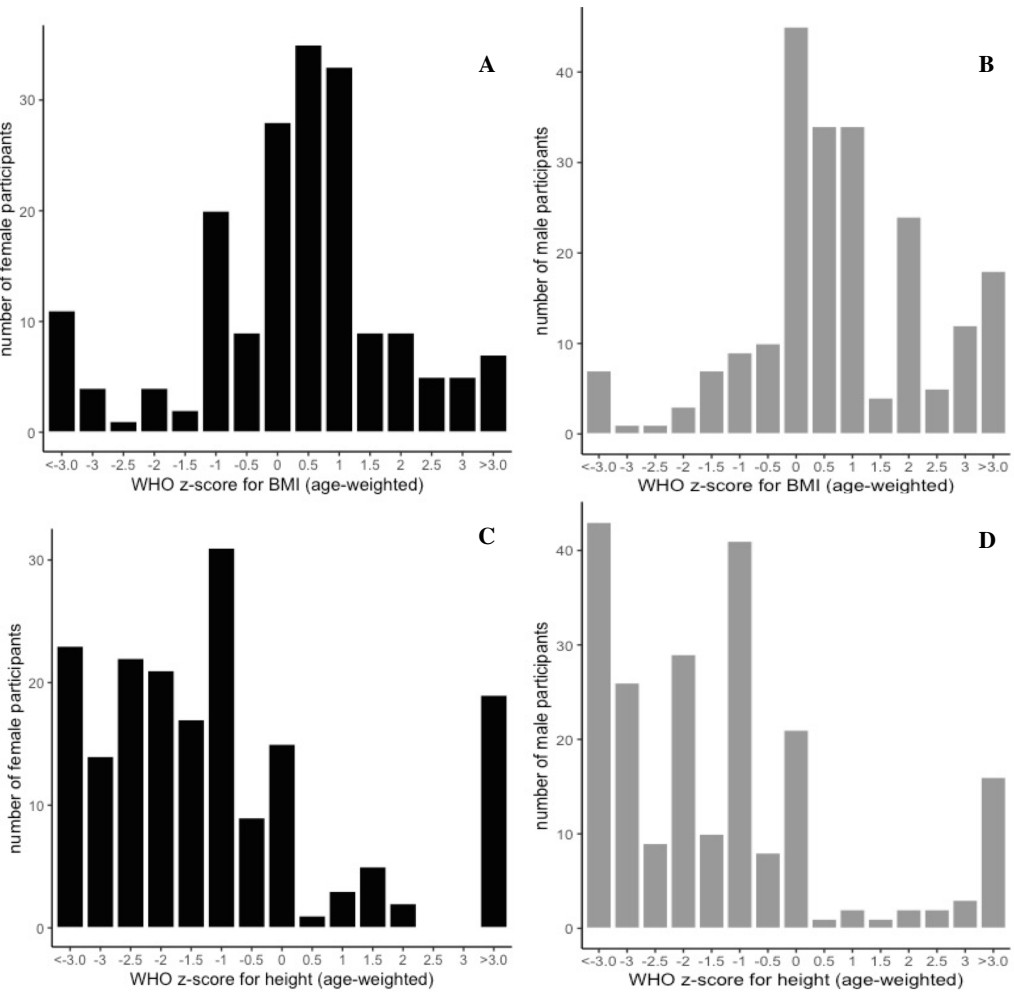

**Figure 2 Gender distribution curves of age-weighted *z*-scores for BMI and height.** Determined according to the WHO guidance (*WHO, 2009*). Grey bars represent male distribution (total male participants, $n = 215$), and black curves represent female distribution (total female participants, n =186), with BMI on the top panel (A, B) and height depicted below (C, D). Male BMI was significantly increased compared to female BMI (t-stat = $-2.4413$, $p = 0.015$), and there was significant variation of height ($p = 0.0000687$), weight ($p = 0.0198$), and BMI ($p = 0.0123$), across communities, by ANOVA, which mirrored significant variation across in age and gender across communities.

## Household water sources and usage

Respondents were asked to disclose all known sources of household water. In Table 1, the percentage of total households in each community reporting each type of water source are illustrated, such that the totals for each community do not add up to 100%, but rather reflect the frequencies of usage of various sources across communities. Although more households were recruited from towns that had a potential lake water source, the number of households using spring water (36.5%) was twice as high as the households reporting lake water usage (15.5%). Interestingly, in addition, 21.3% of households designated bottled water usage and approximately half the houses cited direct municipal water usage (48.6%),

**Table 1   Lake, spring, municipal and bottled water % household reported usage by community.**
Totals do not equal 100% as respondents could select more than one water source. The majority of respondents reported municipal-sourced water, which could not be verified as a lake or spring source or mixed in towns with more than one municipal source because addresses were not taken and surveys were anonymized. Data analysis was based on reported source type and not on an assumed water source per community.

| Location | Lake (%) | Spring (%) | Municipal (%) | Bottled (%) |
|---|---|---|---|---|
| **All** | 15.5 | 36.5 | 48.6 | 21.3 |
| **Agua Escondida** | 0.0 | 82.1 | 78.6 | 7.1 |
| **Panajachel** | 13.3 | 23.3 | 46.7 | 20.0 |
| **Santiago Atitlán** | 26.5 | 0.0 | 23.5 | 50.0 |
| **San Jorge La Laguna** | 0.0 | 56.0 | 38.7 | 6.7 |
| **San Lucas Tolimán** | 35.8 | 20.3 | 49.6 | 29.3 |
| **San Pedro La Laguna** | 6.7 | 8.5 | 76.7 | 28.3 |
| **Tierra Linda** | 3.6 | 83.3 | 16.7 | 0.0 |

**Table 2   Household reported % treatment overall and as distributed within each community.** Household respondents could select as many treatment types that applied. Treatment of any kind * was protective against shedding of *Giardia* (RR = 0.29, $p < 0.0001$, z-stat = 4.42, OR = 0.24).

| Location | Water treatment | Boiling | Cloth filter | Bleach | Clay filter | Carbon filter | Candle | Sand | Unspecified | Any filter |
|---|---|---|---|---|---|---|---|---|---|---|
| **All** | 77.3* | 41.4 | 7.8 | 3.0 | 12.2 | 11.5 | 2.7 | 2.0 | 2.0 | 30.4 |
| **Agua Escondida** | 89.3 | 82.1 | 0.0 | 7.1 | 10.7 | 14.3 | 3.6 | 0.0 | 0.0 | 28.6 |
| **Panajachel** | 80.0 | 40.0 | 0.0 | 10.0 | 0.0 | 33.3 | 0.0 | 0.0 | 3.3 | 36.7 |
| **Santiago Atitlán** | 29.4 | 17.6 | 5.9 | 0.0 | 2.9 | 0.0 | 2.9 | 0.0 | 0.0 | 5.9 |
| **San Jorge La Laguna** | 96.0 | 36.0 | 22.7 | 1.3 | 9.3 | 21.3 | 1.3 | 0.0 | 4.0 | 36.0 |
| **San Lucas Tolimán** | 78.0 | 39.8 | 8.1 | 1.6 | 17.9 | 4.1 | 1.6 | 6.5 | 2.4 | 32.5 |
| **San Pedro La Laguna** | 83.3 | 33.3 | 3.3 | 1.7 | 25.0 | 15.0 | 8.3 | 0.0 | 1.7 | 50.0 |
| **Tierra Linda** | 63.3 | 50.0 | 0.0 | 3.3 | 3.3 | 6.7 | 3.3 | 0.0 | 0.0 | 13.3 |

with some clarifying that as either lake or spring and others uncertain. Other designated sources of water reported included schools and hospitals. Lake water usage varied across municipalities with known lake sources- with San Lucas Tolimán households reporting the highest at 35.8%, followed by Santiago Atitlán (26.5%), Panajachel (13.3%), and San Pedro La Laguna (6.7%). Notably, Santiago de Atitlán reported 50% bottled water usage.

Household water treatment was notable with 77.3% households designating point of use treatment with boiling being the most common (41.4%), followed by various kinds of filters –clay (12.2%), carbon (11.5%), candle (2.7%), sand (2.0%), cloth filter (7.8%), and unknown (2.0%). Treatment with sodium hypochlorite or bleach was the least commonly used technique (3.0%). Table 2 illustrates the breakdown of treatment types across each community. Notably, the lowest proportion of household treatment rate was in Santiago Atitlán (29.4%), with the majority of those using boiling techniques (17.5%), followed by cloth filter (5.9%). San Jorge La Laguna had the highest percentage of households reporting treatment (96%). Not surprisingly, Santiago Atitlán households, with lowest percentages of treatment, were also at higher risk for frequent episodes of diarrhea (OR = 8.78, z-stat
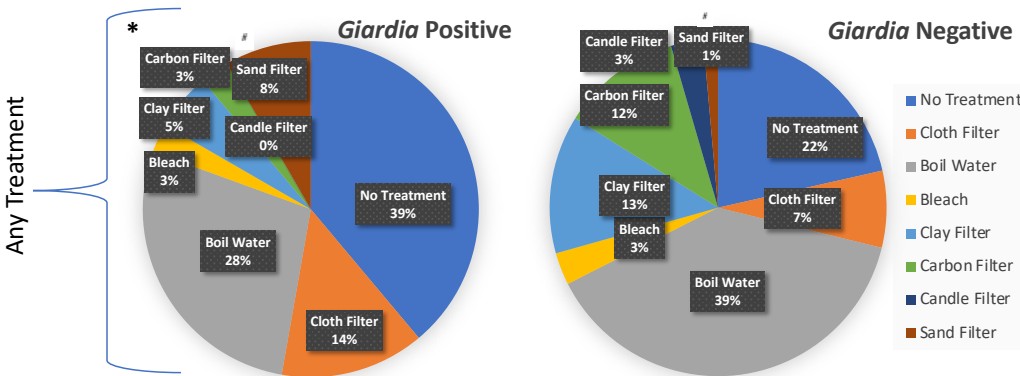

**Figure 3 Household reported % treatment distributed by *Giardia* test outcome.** Logistic regression was used to identify significant characteristics of *Giardia* positive households. Respondents could select as many treatment types that applied. Gravity-fed filters included candle, carbon, ceramic pot, and sand filters. Santiago Atitlán (29.4% for any treatment) and Tierra Linda (63.2% for any treatment) also had a higher percentage of households with frequent episodes of diarrhea and fever (as defined as at least once per month, or greater). Treatment of any kind* was protective against shedding of Giardia (OR = 2.5, z-stat =4.72, $p < 0.008$). #Sand filters increased risk for Giardia (OR = 6.1, z-stat = 2.49, $p < 0.048$). The analysis included a total of 35 Giardia positive samples and 366 Giardia negative samples.

= 4.343, $p < 0.0001$) and fever (OR = 6.39, z-stat = 4.002, $p = 0.0001$), relative to all other households.

Figure 3 illustrates the breakdown of treatment types across *Giardia* negative and *Giardia* positive cases. The presence of any household treatment was protective against *Giardia* presence or absence (OR = 2.5, z-stat = 4.72, $p < 0.008$) (Table 3). When categories of treatment were further broken down and analyzed through logistic regression, the only significance identified was for sand filters (OR = 5.77, z-stat = 2.34, $p = 0.0195$)) with a net harmful effect; however, the limited sample size ($n = 8$) was concentrated in one town (San Lucas Tolimán), giving little confidence in that observation of significant correlation. While there was no significant protection or harm ($p < 0.05$) offered by other individual treatments, our sample size may have not been large enough to tease apart protection or harm for each particular treatment, given that 7.5% of households with treatment, utilized more than one treatment type. A higher proportion of positive *Giardia* cases with use of cloth filters indicates they may provide a false sense of security if not combined with other more effective treatments. These findings suggest that municipal level chlorination combined with household use of gravity -fed filters or boiling could substantially reduce the risk of fecal protozoa for children under 5.

Other uses for household water that could result in routes of ingestion included use of untreated household water for washing of dishes (57.6% overall, with 76.5% respondents in Santiago Atitlán, 85% in San Pedro La Laguna, 66.7% in Panajachel, 57.7% in San Lucas Tolimán, 53.3.% in Tierra Linda, 44.0% in San Jorge La Laguna, and 25% in Agua Escondida) and for rinsing of fruits and vegetables (40.4% overall, with 76.5% respondents in Santiago Atitlán, 53.3.% in Tierra Linda, 51.7% in San Pedro La Laguna, 43.3% in Panajachel, 43.1% in San Lucas, 17.3% in San Jorge La Laguna, and 10.7% in Agua

**Table 3 Demographic and household risk factors for *Giardia* and measures of association.** Odds ratios were calculated for binary logistic regression outcomes and indicate correlation, not causation. Categories listed represent those with significance greater than $p < 0.05$. * indicates the category with increased risk. Three years of age was selected as a cut-off point due to marked turnover of gut microbiota until approximately 36 months [23], with remodeling of the intestinal lining. Height-adjusted $z$-score deviations of greater than 1.5 were examined for increased risk and $z$-scores of less than or equal to $-2.0$ were identified as significant. The analysis included a total of 35 *Giardia* positive samples and 366 *Giardia* negative samples.

| Case Incidences of *Giardia* in Children Presenting with and without Identified Risk Factors | Outcome | | Measures of Association | | | |
|---|---|---|---|---|---|---|
| | *Giardia* positive | | Odds ratio | 95% confidence interval | z- statistic | Significance level |
| *Risk Categories* | Y | N | | | | |
| ≥3 years in age* | 28 | 199 | 3.3568 | 1.4299 to 7.8805 | 2.781 | $p = 0.0054$ |
| <3 years in age | 7 | 167 | | | | |
| height-adj z-score ≤-2* | 23 | 167 | 2.2839 | 1.1032 to 4.7282 | 2.225 | $p = 0.0216$ |
| height-adj z-score >-2 | 12 | 199 | | | | |
| no household treatment* | 14 | 77 | 2.5022 | 1.2161 to 5.1481 | 2.492 | $p = 0.012$ |
| household treatment | 21 | 366 | | | | |

Escondida). Other household uses included watering of plants, bathing, cooking and related activities.

## Sanitation & hygiene

While this study mainly focused on water sources, usage, and treatment as the main risk factors of enterogenic illnesses in this population, we did attempt to gage sanitation conditions in the home. The vast majority of households (92%) reported usage of a latrine, as compared with 3% only reporting a toilet available in the home. The remaining 5% abstained from the question. This trend was consistent across communities; however, there was no increased risk of shedding of *Giardia* with latrine usage at risk for cross traffic and contamination, 40% of households also shared latrines with other houses, and 40% had open access to their latrine. The former did not result in increased risk of *Giardia* shedding among the children in this study, while children in households with open access to the toilet did have 2.0 times the risk of shedding *Giardia* (OR = 2.04, z-stat = 1.992, $p = 0.0464$). There was no increased risk of frequent diarrhea, vomiting, or fever in households with latrines, shared access, or open access.

## Prevalence of *Giardia* and *Cryptosporidium* in children 5 years and younger

Overall, the total prevalence of parasite shedding was 12.2% with a prevalence of *Giardia* shedding of 9.7% (M 9.3%, F 10.8%) and of *Cryptosporidium* shedding of 2.5% (M 0.9%, F 2.2%). A number of risk factors for positive tests for *Giardia* emerged—including increased age, low $z$-score for height and lack of any household water treatment (Table 3). There was increased risk of *Giardia* infection for child ages 3 to 5 as compared with under 3 years of age (OR = 3.36, z-stat = 2.781 $p = 0.0054$), as well as low $z$-score for height (OR = 2.28, z-stat = 2.225, $p = 0.0216$ for those with a $z$-score of $-2$ or less).

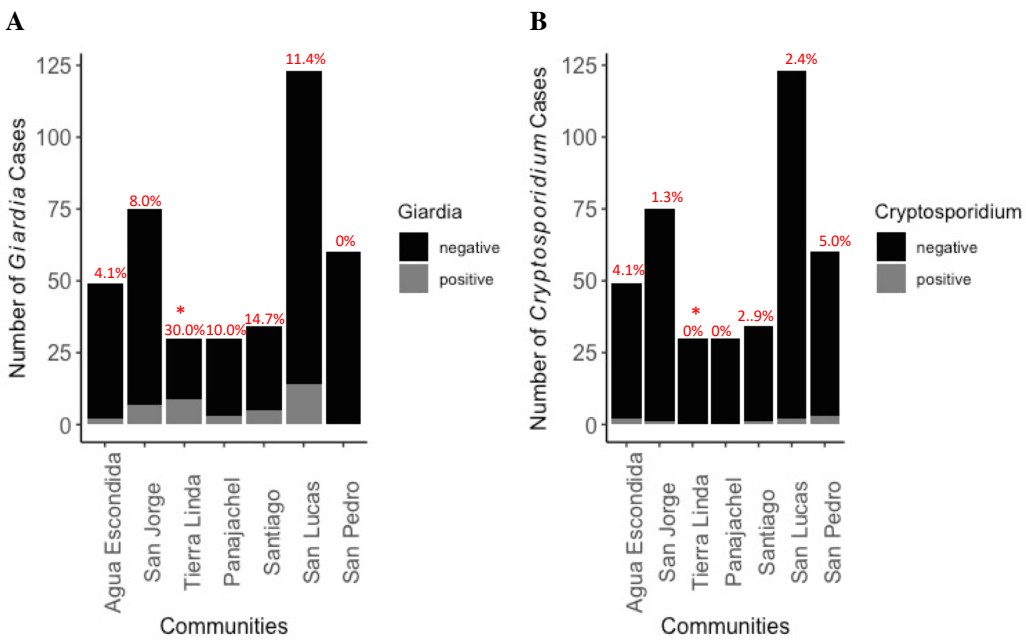

**Figure 4** **Distribution of *Giardia* (A) and *Cryptosporidium* (B) cases by community among total number of participants.** Percent cases by location, indicating community prevalence, is placed atop the bar graph. No cases of comorbidity with both parasites were found. * indicates significantly higher prevalence was found in Tierra Linda compared to other towns ($p < 0.001$). There was 7.8 greater odds of becoming infected with *Giardia* if living in Tierra Linda as compared with other towns (z-stat = 4.48, $p < 0.0001$). The analysis included a total of 35 *Giardia* positive samples and 366 *Giardia* negative samples, as well as 10 *Cryptosporidium* positive and 391 negative samples.

Lack of any form of household treatment of water (OR = 2.5, z-stat = 2.492, $p = 0.012$) posed an increased risk factor for shedding of *Giardia* cysts by the child, both suggesting a direct link with water source and a need to expand and improve outreach around WASH interventions in the region as an effective means to reduce exposures in the household.

Prevalence of shedding of *Giardia* and *Cryptosporidium* by communities is depicted in Fig. 4. Infections with each organism were approximately equally distributed among boys and girls. There were two cases of co-infection with *Entamoeba histolytica* and *Giardia lamblia*, but no cases of co-infection with *Cryptosporidium*. Tierra Linda had a particularly high prevalence, notable for a mountain spring sourced town; indeed, there was a 7.8 greater odds chance of becoming infected with *Giardia* if living in Tierra Linda as compared with other times (OR = 7.8, z-stat = 4.48, $p < 0.0001$). Although other sources of fecal organisms cannot be ruled out, we detected issues with filters in the associated nursery school with health worker outreach so results were affected by contamination of the school water source (Table S3, fecal coliforms were elevated in filtrate). We communicated concerns back to community contacts to ensure filters and containers were decontaminated, cleaned, and replaced properly.

**Table 4 Identified household risk factor for gastrointestinal outcomes and measures of association for binary outcomes of frequent episodes of symptoms reported at household level.** Odd ratio were calculated after binary logistic regression indicated significance ($p < 0.05$). A total of 62 households self-reported lake water usage, and 339 households indicated an alternate source. There were some non-responses for the questions regarding gastrointestinal illness, with a total of 387, 385, and 395 recorded responses for diarrhea, vomiting, and fever, respectively.

| Case Incidences of Gastrointestinal Outcomes in Children in Lake Water and Non- Lake Water Households | Outcome | | | | | |
|---|---|---|---|---|---|---|
| | Frequent Diarrhea | | Frequent Vomiting | | Frequent Fever | |
| Household Groups | Y | N | Y | N | Y | N |
| Exposed to Lake Water | 23 | 39 | 12 | 49 | 18 | 43 |
| Not Exposed to Lake Water * | 62 | 263 | 17 | 307 | 46 | 288 |
| Odds ratio | 2.5 | | 4.4 | | 2.6 | |
| 95% confidence interval | 1.3938 to 4.4900 | | 1.9909 to 9.8243 | | 1.3928 to 4.9315 | |
| z- statistic | 3.073 | | 3.651 | | 3.0 | |
| Significance level | $p = 0.0021$ | | $p = 0.0003$ | | $p = 0.0028$ | |

Notes.
*Indicates not exposed to lake water in household, although may be exposed through other locations Frequent was defined as at least monthly.

## Comparison across kits and with screening by microscope

In comparing the performance of the two purchased ELISA kits, 140 Triage results were compared with the 401 Chek kits. Six of the Triage kits (4.29%) were invalid when run, while only 1 of the Chek produced an invalid result (0.25%). There was very poor correlation between the kits - the Triage kit did not detect *Giardia* in 50% of the samples (11/22) that had a positive test *via* the Chek kit. Seven samples in the kit comparison tested positive for *Cryptosporidium via* the Chek kits; none of these were positive *via* the Triage kits.

Of the 401 samples to submitted to the laboratory, 268 were also examined by fecal flotation and microscope, including 110 of which all had Triage analysis. Five samples had eggs of *Ascaris lumbricoides*, seven had cysts *Iodamoeba butschlii*, seven had cysts of *Blastocystis hominis*, sixteen had cysts *Entamoeba coli* with 1 additional sample of trophozoites (three presented in samples with other organisms), and one sample had *Chilomastix mesnili* trophozoites. Three of these had *G. lamblia* cysts (*A. lumbricoides* in two samples, *E. coli*, and *B. hominis*) and one had co-infection with *Cryptosporidium* (*Entamoeba coli*), but all of these are not necessarily pathogenic. *E. coli, I. butschlii,* and *C. mesnili* are common hominid intestinal inhabitants, while *A. lumbricoides* can cause intestinal blockage and impair growth in children. The pathogenicity of *B. hominis* is unclear.

Six microscope samples identified *Giardia;* the five cysts were also identified by the CHEK kit, while the one trophozoite positive sample was not. Only one of the cyst samples was identified by the Triage kit.

## Household reported illness in children 5 years and younger

Household lake water use emerged as a risk factor, among household characteristics for frequently reported gastrointestinal symptoms, *e.g.*, at the very least several times per month, (OR = 2.5, z- stat = 3.07, $p = 0.0021$ for diarrhea; OR = 4.4, z-stat = 3.65, $p = 0.0003$ for vomiting; and OR = 2.6, z-stat = 3.0, $p = 0.0028$ for fever), despite not

being a predictor of shedding of protozoa in this study. Table 4 provides totals for each separate outcome in exposed and unexposed groups, the odds ratio, and other measures of association for binary logistic regression, for each separate potential gastrointestinal symptom or outcome. In households with lake water use, children were twice as likely to have frequent episodes of diarrhea with 40% of households utilizing lake water reporting diarrhea at least once a month as compared with 16% for non-lake sourced households. Similar trends were observed for fever (28% *versus* 11%) and vomiting (19% *versus* 7%).

Figures 5A–5C depict the full distribution self-reported frequency of diarrhea (A), vomiting (B), and fever (C) across communities, with three levels of frequency: symptom reported frequently (monthly or greater), sometimes (annually or greater), and never reported. There was also an increased risk by location for symptoms. Santiago Atitlán had an increased risk for both diarrhea (OR = 8.8, z-stat = 4.343, $p < 0.0001$) and fever (OR = 6.4, z-stat = 4.002, $p = 0.0001$), compared with other locations. Tierra Linda also had an increased risk of diarrhea (OR = 2.5, z-stat = 2.155, $p = 0.0311$) in children 5 years and younger, compared to other locations. As can be seen in Figs. 5A–5C, the vast majority of households across communities reported diarrhea and fever at least several times per year, and approximately half the households reported vomiting at least several times per year.

## DISCUSSION

This study substantially contributes to the dearth of published works characterizing gastrointestinal illness among indigenous lakeside communities at Atitlán. It reveals a strong association of household use of water and reporting of frequent adverse gastrointestinal symptoms of diarrhea, vomiting (Table 4). Profound to note was the frequency all three gastrointestinal symptoms were reported greater than monthly and at least several times per year, across all locations. In contrast, there was no noticeable increased risk with self-reported sanitation patterns (*e.g.*, type of household sanitation and degree of access), although 40% of households reported open or shared access. A major limitation of the study was the inability to simultaneously assess a wide range of enteric pathogens, including viruses such as rotavirus, or bacteria such as cholera or salmonella. The lower than expected infection rates of common protozoal diseases (12.2%) cannot sufficiently explain the high prevalence of 40% of more than monthly episodes of diarrheal disease in this region. This prevalence corroborates findings and rates reported by the Ministry of Health (*INE, 2018*). These results also potentially represent a better resourced portion of the indigenous population at the lake, *e.g.*, those seeking out health services or connected to existing programs. While there was no direct connection between source of household water as a risk factor and shedding of parasites, there was a substantial increased risk of frequent gastrointestinal illness in children 5 years and younger from household use of lake water (odds ratio risk of 2.5, 4.4, and 2.6; *p*-values of 0.0021, 0.0003, and 0.0028, for diarrhea, vomiting, and fever, respectively), suggesting the importance of addressing lake water quality as a major risk factor for the health of children at Lake Atitlán. In addition, households with open access to latrines had an increased risk of *Giardia* shedding among the children in this study (OR = 2.04, z-stat =1.992, $p = 0.0464$), suggesting sanitation practices and hygiene could be central to reducing exposure to fecal protozoa.

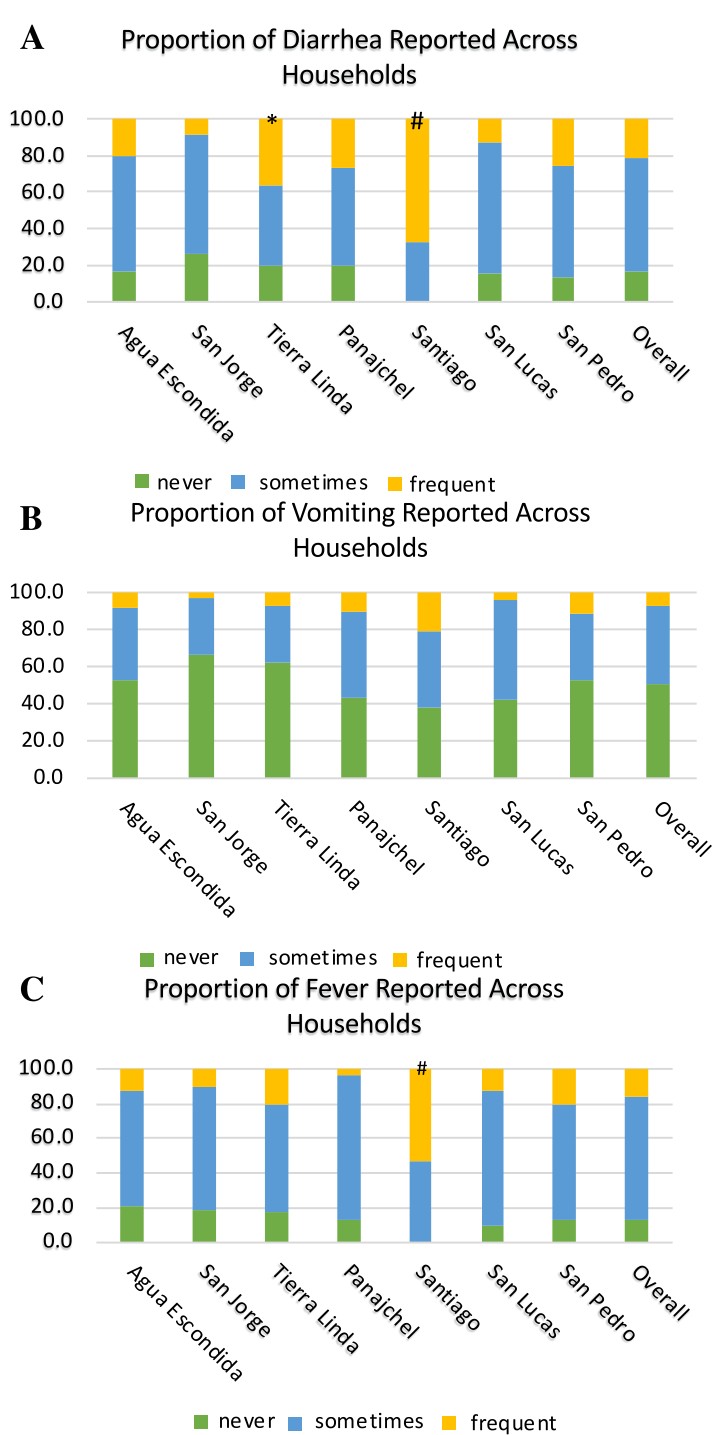

**Figure 5** **Proportions of households reporting diarrhea (A), vomiting (B), and fever (C) across locations.** The legend denotes the proportion of household reporting never, sometimes, and frequent (greater than once a month) episodes of each symptom in children five years and younger in each household. # denotes significance of $p \leq 0.001$ of risk compared to other locations. * denotes significance of $< 0.05$ compared to other locations. Santiago Atitlán had 8.8 times the risk for diarrhea (z-stat $= 4.343$, $p < 0.0001$) and 6.4 for fever (z-stat $= 4.002$, $p = 0.0001$) in children 5 years and younger. Tierra Linda had an increased risk 2.5 for diarrhea (z-stat $= 2.155$, $p = 0.0311$), compared to other locations. Sample sizes of self-reporting for diarrhea, vomiting, and fever were 387, 385, and 395, respectively.

Our rates of *Giardia* and *Cryptosporidium* were lower than previously reported in the region. Close to 50% in preschoolers in three Guatemalan Western highland pre-schools were actively shedding fecal protozoa and 43% of children had multiple parasitic infections (*Jensen et al., 2009*). 32% of children in villages around Atitlán presenting with gastrointestinal symptoms were shedding *Cryptosporidium oocysts* (*Bentley et al., 2004*; *Laubach et al., 2014*). A prevalence of 21.5% for *Giardia duodenalis* and 19.8% for *Entamoeba histolytica* was determined after fecal samples were collected to field visits to clinics in the highlands and riparian zones (*Soto-Méndez et al., 2016*). It is important to note that our study examined mostly "healthy" children. Those not presenting with symptoms at the time of fecal sampling may have exhibited a greater association of protozoal shedding with lake water use if sampled when symptomatic.

Previous work comparing the Triage Micro Parasite Panel with traditional ovum and parasite microscopy (O & P) for the detection of *G. lamblia, C. parvum,* and *E. histolytica* indicated a much higher sensitivity and specificity than observed in this study. When utilized to screen known positive and negative stool samples with O & P, including trichrome and modified acid-fast stains, the sensitivity and specificity of the test for listed pathogens was 95.8% and 97.4%; 98.3% and 99.7%; and 96.0% and 99.0%, respectively (*Garcia, Shimizu & Bernard, 2000*). In this study, a number of samples with discrepancies with positive Triage samples were found to be true positives when tested by an alternative method of immunofluorescence or enzyme immunoassay (EIA) panel. We note, following the manufacturer's instructions for sample preparation, the dilution of the Triage sample was double that of the final Chek kit sample. The underperformance of the Triage kit could be related to a lower concentration of parasite, particularly in asymptomatic individuals. Indeed, in symptomatic and ill patients, testing negative by microscopy alone, (*Sharp et al., 2001*) and in outbreak scenarios (*Swierczewski et al., 2012*), where fecal shedding and load individuals is high, and when prevalence of disease is high, these rapid bedside kits may be particularly useful tools in tagging samples for further analysis beyond O & P. However, in our study, of the 17 *Giardia* positive cases detected by Chek kit, only 29.4% were positive by O & P, indicated a utility for this test in tiered screening, particularly when individuals are asymptomatic at the moment or when overall suspected prevalence is lower. Previous work comparing detection in known positive and negative samples by the Chek kit platform, along with a number of other enzyme linked or rapid PCR tests, indicated a sensitivity and specificity of 100%, 94%, and 100% and 100%, for *G, lamblia* and *C parvum, re* these (*Van den Bossche et al., 2015*); an alternative work compared the test to microscopy in 200 stool samples from clinical cases and found 93.3% and 99.4%; 87.6% and 98.9%, respectively for sensitivity and specificity of these two organisms (*Alexander, Niebel & Jones, 2013*). Our results here indicate that in routine surveillance and screening of asymptomatic individuals the Chek kit may provide a more robust platform compared to traditional O & P microscopy, not just the advantage of time and decreased level of skill. Indeed, the Chek assay outperformed quantitative polymerase chain reaction as a rapid point-of-care diagnostic for infants in Bangladesh (*Kabir et al., 2018*).

Our study sample size and distribution had some limitations in examining potential differences between communities as demonstrated by the physical exam data. Recruitment

was not necessarily reflective of population size of the respective town. Santiago Atitlán, Panajachel, and San Lucas Tolimán have some of the larger and most concentrated populations in the region (estimated between 20–50K) (*Ferráns et al., 2018*), depending on exact boundaries and time of the year, while the other villages are notably smaller. Our recruitment was directly through community partners and health care workers. This permitted greater participant trust and retention, as well as the confidential distribution of results and follow up care. Previous community-based work has demonstrated direct engagement with households around filter use or other intervention methods can drastically improve water quality at point of use in the household (*Roegner et al., 2016*). However, this approach has its limitations and sampling bias as we had uneven participant distribution among locations (Fig. 1). Additionally, the sampling was biased toward individuals already engaged within programs that address or advocate WASH and/or means to seek out health services (*Omarova et al., 2018*; *Setty et al., 2019*).

Household treatment was reported at higher levels than previously estimated (*Laubach et al., 2014*; *MSPAS, 2019*; *Nagata et al., 2011a*; *Nagata et al., 2011b*). This discrepancy may reflect the extent of outreach by health care provider facilities and NGOS, through which the project was implemented. Our education handouts included considerations about risk through other avenues (Appendix A), and future outreach work in the region should incorporate recommendations beyond exposure directly at the tap.

It is also important to acknowledge that lake water usage was lower than anticipated among lake-sourced towns (in particular, San Pedro La Laguna, Santiago Atitlán, and San Lucas Tolimán). This study did not evaluate parent or guardian knowledge and attitudes regarding lake water safety. It is unclear whether the lower levels of lake water usage reflects households as viewing lake water as 'unsafe' and seeking out other sources (for example, we note the high use of bottled water), lack of information about from where municipal water is actually derived, or truly lower levels of lake water distribution in towns. A study conducted in Santiago Atitlán regarding the drinking water beliefs showed that more than half of the participants considered the lake's water to be dirty due to the contaminants that entered the lake (*Nagata et al., 2011a*; *Nagata et al., 2011b*). The sources of contamination included trash, rain, bacteria and germs, laundry run-off and boats. Even though historically people have mistrusted chlorinated water, a slight majority of the participants preferred drinking chlorinated water as they considered it could help kill bacteria and germs, clean the water, prevent illness and prevent diarrhea. In comparison, those who did not like chlorinated water mentioned they disliked the taste and smell, they considered it could cause illness, it was not effective and that it was too strong (*Nagata et al., 2011a*; *Nagata et al., 2011b*).

Thus, a major limitation of our study was reliance on self-reporting of water source. We lacked the ability to geo-reference household sites with municipal officials to confirm exact water source piped to the household. Both for confidentiality purposes and uncertainty of validity of georeferencing, this additional tracing step was omitted from the study, but future work is needed to (1) delineate and verify water sources for neighborhoods and households from these lakeside towns and (2) capture perceptions in community about safety of various water sources. A higher than expected level of household lake water usage was found in Panajachel. With known high levels of arsenic in the well water

in Panajachel, we are uncertain as to whether behavioral changes have occurred due to viewing spring water as 'unsafe,' lack of knowledge about source of municipal water, or whether this reflects a true trend. Similarly, the simultaneous high usage of bottled water in Santiago Atitlán, along with the lowest levels of household water treatment, and the higher reported frequency of gastrointestinal illness in children under 5 years of age, lends serious concern to the potential false sense of security and perceptions about the safety of locally bottled water sources. Future work must include testing of local bottled water sources and education campaigns about precautions with uncertified labels.

It is also very likely that other household WASH practices play a large role in rates of protozoal infections, and broad WASH best practices must be further targeted to further reduce the incidence. Indeed, households with open access to latrines had an increased risk of *Giardia* shedding among the children in this study (OR = 2.04, z-stat = 1.992, $p = 0.0464$), suggesting sanitation practices and hygiene could be targeted to reduce this risk. Alternative passive sampling means, such as collection from home latrines (*LaHue & Alexander, 2018*), may more accurately capture exposure and incidence in the home environment, but it would obfuscate access to studying exposure in children 5 years of age and younger. Regardless of route of exposure, if protozoal infections persist, malabsorption of nutrients may occur and result in stunting (*Mmbaga & Houpt, 2017*; *Mohammadkhah et al., 2018*; *Solomons et al., 2015*). Possible explanations for increased risk among participants 3 years of age and older (OR = 3.4, z-stat = 3.12, $p = 0.0058$) include increased risk from direct ingestion of water from tap without treatment (as opposed through nursing or formula prepared with boiling water), increased risk of exposure through fecal-oral routes or other household sources as the child becomes more mobile, exposure to other multiple water sources outside the household, or the pathophysiology of *Giardia* infections. The risk for stunting and developmental growth necessitates a closer look at the incidence of infection rates by multiple organisms, not just shedding of specific protozoa, among children under 5 years of age (*Certad et al., 2017*; *Mmbaga & Houpt, 2017*; *Mohammadkhah et al., 2018*). Furthermore, animals (cats, dogs, chickens) were widely reported with access to the porch or house, indicating a need to further investigate into zoonotic illnesses (*Budge et al., 2019*; *Ercumen et al., 2017*). The methods used in this study focused on *Giardia, Cryptosporidium,* and *Entamoeba* species most commonly carried by humans, but other species can easily infect children and vulnerable groups.

Much lower rates of household water treatment in the region have been previously reported (*Nagata et al., 2011a*; *Nagata et al., 2011b*; *Roegner et al., 2016*). While the results of reported household treatment varied substantially by community, we found that (1) communities and households without additional household level treatment are at increased risk of protozoal infections in children 5 years of age and younger; and (2) communities with lower rates of household treatment of water also report higher incidence of household gastrointestinal illness in children 5 years of age and younger. These findings emphasize the importance of both dissemination of filters for household use and education about proper filter care and use, as well as distribution of chlorination and boiling protocols. Furthermore, it should be emphasized that cloth filters are insufficient treatment methods

on their own. Finally, these household approaches combined with enforcement of state-mandated chlorination treatment at municipal plants could go a long way in reducing both prevalence of childhood protozoal infections and gastrointestinal illness around the lake. While beyond the scope of this study, future work could more closely examine the relative efficacy of various treatment practices individually and together in the context of the household longitudinally in reduction of pathogens, including protozoa, bacteria, and viruses, and prevention of gastrointestinal illness outcomes, particularly in young children.

Finally, sewage inflow into the lake is largely untreated with direct inflow at Panajachel, Santa Catarina Palopó, and Santa Cruz La Laguna, with high levels of nitrogen and phosphorus entering and only minimal treatment at other locations, thereby still contributing substantial nutrient loads (*Chandra et al., 2013*). Rivers San Francisco and Quiscab, surrounding Panajachel to the east and west, also drain the basin bringing wastewater discharged in the upper highlands into the lake, as well. Previous work in the region has documented *Escherichia coli* at lake water at these outflows at 1,800 and 10,000 MPN/100 mL, respectively. Similar values were observed at various beaches and lake sites routinely frequented for recreation (*Roegner et al., 2016*). In the same study at San Pedro La Laguna, household tap water, prior to treatment, had levels of fecal coliforms and *E. coli,* exceeding permissible levels by the World Health Organization. The local governmental group Authority for the Sustainable Management of the Lake Atitlan Basin and its Surroundings (AMSCLAE) has recently reported similarly elevated levels of coliform at other sites around the lake (*Ferráns et al., 2018*), and historical data also supports lake wide contamination at intake, point sources, and nonpoint sources (*Castellanos et al., 2002*). While beyond the scope of this study to evaluate pathogens in source waters and efficacy of any employed treatment, we did take single time point samples for each municipality at the drinking water intake from the lake or mountain source and then from a tap source, post variable levels of treatment (Table S3). Levels were typically lower at the intake than previously reported testing near discharge sites. However, levels in untreated lake waters were typically indicative of a moderate to severe risk for adverse effects based on WHO guidelines for recreational use. Treated tap water taken from the lake were typically orders of magnitude lower in risk, further supporting the reduced risk in the presence of household treatment found this study. Given the limited time points and samples, we did not include these findings in our overall presentation of this work. Yet, overall, it is clear that widespread wastewater discharge in the region can be directly linked to declining source water quality with severe implications for the gastrointestinal health and development of children in the region. Not only is adequate treatment and infrastructure for drinking water needed to adequately augment children's health and development in the region, the underlying issue of wastewater contamination and discharge into the lake must be addressed.

## CONCLUSIONS

Our study contributes to a better understanding of the gastrointestinal health burden in young children from indigenous communities around Lake Atitlán, Sololá, Guatemala.

Our results underscore the relationship to water quality, and provide potential areas for implementing WASH interventions and educational outreach in the region to better meet indigenous community needs. Key findings for public health officials and water resource managers include:

- Improved household level treatment appears to reduce risk, but is not uniformly distributed.
- Lake water, untreated water usage, and contaminated bottled water are all key points of risk to address, particularly with respect to reducing adverse gastrointestinal health outcomes in children 5 years and younger.
- Both treatment and lower infection rates may be misrepresented in our population already engaged through existing health provider or NGO avenues.
- The prevalence of protozoal infections cannot account for the incidence and frequency of gastrointestinal illness in the region, and other enteropathogens must be examined in future studies.
- Although incidence of shedding of *Giardia* and *Cryptosporidium* was lower than previously reported, it still represents a substantial burden with implications for malabsorption, stunting, and other chronic developmental effects, particularly in otherwise asymptomatic children.
- Additional education is needed about other avenues of contamination, including but not limited to rinsing plates or produce with untreated water, risk from animals, and sanitation risks from open latrine access.
- The ELISA Chek Kit utilized in this study appears to be a useful tool for quick and straightforward diagnostic evaluation of *Giardia lamblia* and *Cryptosporidium parvum* in feces in the region, particularly in patients not currently presenting with symptoms
- Future investigations should explore the risk from locally bottled water, perceptions that different households have with respect to drinking water risk, and how that influences behaviors or choices. In addition, other pathogens of fecal origin, including bacteria and viruses should be examined with the relationship of risk to water quality and source.

## ACKNOWLEDGEMENTS

We thank the Rotary Club at Lake Atitlán, Panajachel, for donating the anti-protozoal medications used to treat Giardia-positive children under the supervision of a healthcare provider. We also thank the Association of Amigos del Lago de Atitlán, and Asociación Vivamos Mejor Guatemala for logistical and personnel support, as well as Laboratorio La Asunción for help with diagnostics and screening by microscope. Finally, the project would not have been possible without the enthusiastic involvement of Audrey Ward (Medical Program Coordinator) and Mayan Families staff and volunteers; Dr. Anita Tuch and the San Pedro Centro de Salud medical staff; Dr. Juan Manuel Chuc and the Hospitalito de Atitlán medical staff; Dr. Rafael Tun and the San Lucas Tolimán Hospital medical staff—their cooperation, coordination, and care are greatly appreciated.

### Funding

Funding for this project was made possible by a Seed Grant for International Activities from the University of California, Davis. The funders had no role in study design, data collection and analysis, decision to publish, or preparation of the manuscript.

### Grant Disclosures

The following grant information was disclosed by the authors:
International Activities from the University of California, Davis.

### Competing Interests

The authors declare there are no competing interests.

### Author Contributions

- Amber Roegner conceived and designed the experiments, performed the experiments, analyzed the data, prepared figures and/or tables, authored or reviewed drafts of the paper, and approved the final draft.
- Mónica N. Orozco conceived and designed the experiments, performed the experiments, analyzed the data, authored or reviewed drafts of the paper, and approved the final draft.
- Claudia Jarquin conceived and designed the experiments, authored or reviewed drafts of the paper, and approved the final draft.
- William Boegel conceived and designed the experiments, performed the experiments, authored or reviewed drafts of the paper, and approved the final draft.
- Clara Secaira and Marlin E. Caballeros performed the experiments, analyzed the data, authored or reviewed drafts of the paper, and approved the final draft.
- Lujain Al-Saleh performed the experiments, prepared figures and/or tables, and approved the final draft.
- Eliška Rejmánková conceived and designed the experiments, performed the experiments, analyzed the data, authored or reviewed drafts of the paper, and approved the final draft.

### Human Ethics

The following information was supplied relating to ethical approvals (i.e., approving body and any reference numbers):

Human subjects approval and work with sensitive subjects (children) was approved through the Institutional Review Board at University of California, Davis (IRB # 1070493-3) and through the Comité de Ética en Investigación of the Centro de Estudios en Salud (CEI-CES) at Universidad del Valle de Guatemala (No. Protocol 167-07-2017).

### Data Availability

The raw data is available in the Supplementary File.

Supplemental Information

Supplemental information for this article can be found online at http://dx.doi.org/10.7717/
peerj.12331#supplemental-information.

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
