# Peer review of "Childhood parasitic infections and gastrointestinal illness in indigenous communities at Lake Atitlán, Guatemala"

_PeerJ, doi:10.7717/peerj.12331_

## Round 0.1 · original submission · Minor Revisions

Your manuscript is reviewed by 6 reviewers. They have provided some suggestions to improve the manuscript. Please make the changes in the manuscript or answer the reviewer's comments.

·

Basic reporting

Amber Roegner et al., the article entitled "Childhood parasitic infections and gastrointestinal illness in indigenous communities at Lake Atitlán, Guatemala," is a very well conducted and written study.

Experimental design

Appropriate as per study design, no further comments

Validity of the findings

Authors estimate the prevalence of shedding of waterborne fecal parasites among children 5 years of age and younger, assess the household-reported incidence of gastrointestinal illness in children, and finally characterize water sources, treatment, and
sanitation conditions in households.

They concluded that the prevalence of protozoa could not explain the frequency of gastrointestinal illness, and risk from other enteropathogens must be explored.

Additional comments

Figure quality needs to be improved, significance mark put at appropriate in figures.

Reviewer 2 ·

Basic reporting

No comments

Experimental design

No comments

Validity of the findings

No comments

Additional comments

Amber Roegner et al in their manuscript “Childhood parasitic infections and gastrointestinal illness in indigenous communities at Lake Atitlán, Guatemala “did a cross-sectional study to determine whether household use of untreated lake water results in increased risk of shedding of parasites and gastrointestinal symptoms. Overall, the study is well designed, thoughtfully carried out, and is of prime importance for the community.
I have few minor comments:
In all the figure legend, authors should include the total number of subjects (n), statistical test used and show p values.
The quality of figure 3 ,5 2B, D should improve.

Reviewer 3 ·

Basic reporting

1. Authors might consider making sentences short as it may help in providing a much clearer message to the readers.
2. Graph in Figure 3 appears to be a little confusing on the end highlighting the positive cases of Giardia. The data can be represented in a more suitable way to highlight a positive correlation between the incidence of infection and the respective treatment adopted.

Experimental design

1. Authors have meticulously planned the study and credit should be given to them for honestly highlighting the pitfalls of the study design and the lacunae. Having said that, the authors have incorporated maximum assessable parameters that could influence the outcome of the study.

Validity of the findings

1. It will be interesting to estimate the parasite/bacterial load in the lake at different geographical points that were selected in the study.
2. The microbial load might also be influenced by the hygiene practices adopted by the community. It will be nice to bring forward that correlation if any with respect to the contamination observed in each of the geographical regions.
3. It might not be feasible, but out of scientific curiosity did the authors follow up the microbial load after water treatment strategies adopted by the population?
4. Did the physical wellness inclusion criteria include the possibility of episodic cases of GI symptoms?

Reviewer 4 ·

Basic reporting

no comments

Experimental design

I find an experimental design met all the standard, except the range of pathogens could have been studied in ELISA.

Validity of the findings

I find the results support the conclusion, the study draws attention to waterborne diseases among children and presents the practices that need to follow by the government and community to prevent such outbreaks.

Additional comments

The manuscript is very well written. I also find the use of statistical tests is appropriate and follows the practice of good study design. I recommend the manuscript be accepted in its current form.

Reviewer 5 ·

Basic reporting

The text of the manuscript was clearly written. There are a few minor comments that could improve sentence clarity which are described in the general comments to the author.

The author's also provided enough background on the subject matter and study region for the reader to be able to interpret the findings of the manuscript.

The author's also formatted most of the table and figures correctly. However, Tables 3 and 4 both appear to have two separate tables. I would suggest changing the format of each table and or make them officially 4 tables.

Experimental design

No comment

Validity of the findings

The author's properly evaluated the findings. I would, however, suggest to only report one effect measure from the logistic regression. The author's currently report RR and OR, but choosing just one measure would likely make the manuscript more clear and concise.

Additional comments

As outlined by the authors, the study aims to better understand the connection between water sources used for drinking and other purposes and the health of the communities bordering Lake Atitlan, in Guatemala. Specifically, the authors used fecal samples and household surveys to better understand certain risk factors in the population and then relate this to WASH practices. The authors found the prevalence of shedding Giardia and Cryptosporidium in the population <5 years of age (n = 401) was ~ 11%, but that reports of health symptoms associated with poor water quality was much higher, indicating other factors at play in regards to community health, which provides good grounds for further research.

While there are some aspects of the manuscript which should be corrected or further addressed, I recommend this work to be accepted for publication with minor modifications as outline below.

General: The author’s have a brief paragraph about sanitation and hygiene, but was there any attempt to correlate sanitation with levels of pathogens observed in the fecal samples? If so, this would be an important component of the work. If not, consider removing the sanitation section from the results and just mention this in the discussion to make the manuscript more focused on the analysis.

General: The author’s present both risk ratios (RR) and odds ratios (OR) from some of the analysis. The reviewer has not seen this done before and is unclear if this is appropriate. For example, with logistic regression, the OR is typically the effect measure reported, but under some situations the RR may be a more appropriate measure. Please consider exploring this further and potentially only report one of the effect measures to make the interpretation for concise and clear.

General: The author’s compare two types of test kits to detect Giardia and Cryptosporidium, with some interesting results. Can the authors provide more information about the tests to help the reader interpret the findings that some tests seem to be more/less sensitive? For example, are the tests using the same sample volumes? It would be good to clarify why one might expect differences between tests and the authors might be able to make some recommendations about why certain tests would be expected to perform better under certain conditions and this be an important note for future research.

Specific comments:

Line 88: Please consider clarifying the sentence. Is this just describing the demographics of children <5? Also, a bit confusing that the sentence states these numbers were in 2017, but also 2018 and 2019?

Line 105: Please consider changing “mom” to “mother”.

Line 203: Please consider keeping this in the previous paragraph and adding a citation for float and smear approach.

Line 206: Please clarify what is meant by “banked”, is this the same as archived?

Line 275: The authors mention sand filters in the proceeding sentences being protective again Giardia detection, however, this method treatment does not appear in Figure 3, please correct. Also, please consider adding sample sizes to each group shown in Figure 3.


Figure 4: Can the authors provide information about the population of these different villages? For example, is SL much larger than the other population and would some effort to standardize the data by sample size help with interpretation?

Line 392: Please remove the italics of protozoal.

Reviewer 6 ·

Basic reporting

Roegner and colleagues have studied the impact of direct water sources and different types of water purification methods on the occurrence of water-borne diseases like giardiasis and development of symptoms like diarrhea, vomiting, fever, etc in the inhabitants around the Lake Atitlan, Guatemala. This study becomes important as it provides insights into the most efficient method among different methods used in these communities to purify water that leads to minimum cases of illness. This study could be used to raise awareness in the communities to not use the water directly from natural sources and why filtration is necessary prior to water consumption. The manuscript is well drafted and methods and results are explained in detail. I recommend that the manuscript can be accepted after a minor revision. Below is my comment;
1. Please include the data for prevalence of Cryptosporidium in Children (Figure-4).

Experimental design

The experimental design used by the authors is comprehensive. The study includes the full details of participants like age, BMI, etc. All the methods used in the study are described in full details.

Validity of the findings

All underlying data is being provided except for the data for prevalence of Cryptosporidium in Children (Figure-4). Please include the data.

Additional comments

Roegner and colleagues have studied the impact of direct water sources and different types of water purification methods on the occurrence of water-borne diseases like giardiasis and development of symptoms like diarrhea, vomiting, fever, etc in the inhabitants around the Lake Atitlan, Guatemala. This study becomes important as it provides insights into the most efficient method among different methods used in these communities to purify water that leads to minimum cases of illness. This study could be used to raise awareness in the communities to not use the water directly from natural sources and why filtration is necessary prior to water consumption. The manuscript is well drafted and methods and results are explained in detail. I recommend that the manuscript can be accepted after a minor revision. Below is my comment;
1. Please include the data for prevalence of Cryptosporidium in Children (Figure-4).

---

## Round 0.2 · accepted · Accept

Congratulations, based on reviewers comments I recommend acceptance of your work in PeerJ.

Reviewer 2 ·

Basic reporting

No comment

Experimental design

No comment

Validity of the findings

No comment

Additional comments

Amber Roegner et al in their revised manuscript "Childhood parasitic infections and gastrointestinal illness in indigenous communities at Lake Atitlán, Guatemala" made required/suggested modifications. I am happy with the revised manuscript, revised version is much more improved and it will be more helpful and informative for readers.

Reviewer 3 ·

Basic reporting

NA

Experimental design

NA

Validity of the findings

NA

Additional comments

The authors have addressed the comments satisfactorily.

Reviewer 6 ·

Basic reporting

The authors have revised the manuscript according to my comment and have included the data in the manuscript.

Experimental design

Experimental design is comprehensive.

Validity of the findings

The data is statistically sound and the study is impactful.